# Is There a Correlation between Gingival Display and Incisal Inclination in a Gummy Smile? Study on Cephalometric Parameters

**DOI:** 10.3390/healthcare11030344

**Published:** 2023-01-25

**Authors:** Alessandra Impellizzeri, Raissa Palmigiani, Martina Horodynski, Tiziana D’alfonso, Antonella Polimeni, Adriana De Stefano, Gabriella Galluccio

**Affiliations:** 1Department of Oral and Maxillofacial Sciences, Sapienza University of Rome, 00161 Rome, Italy; 2Department of Computer, Control and Management Engineering, Sapienza University of Rome, 00161 Rome, Italy

**Keywords:** gummy smile, gingival display, incisal inclination, cephalometric parameters, orthodontics

## Abstract

Background: Excessive gingival display or “gummy smile” is a clinical condition where a maxillary gum shows between the inferior line of the superior lip and the gingival line of the incisive superior during a spontaneous smile. The aim of this research was to understand the various skeletal and dentoalveolar components contributing to a gummy smile in a sample of 120 patients. Material and Methods: This retrospective case-control study had the primary objectives of analyzing the existence of a correlation between the presence of gingival exposure and the alteration of the inclination of the upper incisors with respect to the Frankfurt plane, the Palatine plane (bi-spinal) and to the NA line in a sample of orthodontic patients, and also evaluating the association with skeletal, dental, and aesthetic cephalometric parameters. Result and Conclusions: In our study, it’s emerged a correlation between the gingival exposure and the presence of alterations to incisal torque in the vestibular direction and the quantity of maxillary gingiva evident during the smile, which is correlated in particular to the Is–Sts distance, overjet and overbite. The major indicative data, therefore, are related to the vertical position of the upper incisors, in particular with respect to the upper lip and to the sagittal position.

## 1. Introduction

Gummy smile (GS) can be due to excessive vertical bone growth, dento-alveolar extrusion, short upper lip, upper lip hyperactivity or altered passive eruption (APE) [1,2,3].

The GS clinical diagnosis involves a systematic examination of the aesthetics of the face and teeth, which must be performed based on the following three aspects:Analysis of the proportions of the face in the three planes of space with the aim of identifying the facial type, the possible presence of asymmetries, excessive or insufficient facial height and mandibular or maxillary deficit or excess [4];Analysis of the teeth with respect to the face with the aim of quantifying the visualization of the teeth at rest, during speech and smiling, excessive gingival exposure, inadequate exposure of the anterior teeth, inappropriate gingival heights and excessive or deficient corridors buccal [5,6,7];Analysis of the relationship of the teeth between them with the aim of evaluating the dental proportions in height and width, the shape and the gingival contour, the connectors and the incisors [5,6,7,8,9].

For the assessment of gingival exposure, alongside the clinical examination, a photometric detection may be useful, integrating photographs with measurements from cephalometric points. Photometric detection depicts the patient in front and side views with the postural attitude of the upper and lower lips in two circumstances that are easily reproducible: the position of the lips at rest and the position of the lips during the smile [10,11,12,13]. The present study investigated the various skeletal, dentoalveolar and soft-tissue-related components contributing to a GS in a sample of 120 patients, with particular attention to the possible relationship with alteration of the upper incisors’ torque.

## 2. Material and Methods

The present study investigated the various skeletal, dentoalveolar and soft-tissue-related components contributing to a GS in a sample of 120 patients, with particular attention to the possible relationship with alteration of the upper incisors’ torque.

The study was carried out on patients referred to the Orthodontics UOC of the Department of Odontostomatological and Maxillo-Facial Sciences of the “Sapienza” University of Rome. The recruitment period of the patients was 4 years, from 2014 to 2018. The sample patients were formed from 120 total patients for the study and control group, with a range between 7 and 35 years of age. Patients were included in the study group after an evaluation of photographic documentation performed before the treatment commenced, and based on previous studies reporting the acceptable amount of exposed gingiva during smiling, we considered at least 2 mm of gingival exposure during a posed smile as an inclusion criterion.

The inclusion criteria considered in the study design were as follows:Patients affected by gummy smile, with 2 mm of evident maxillary gum during full smile and no spontaneous in extra-oral profile photo;Patients aged between 7 and 35 years;Patients who would be reliable for follow-up;Patients with no previous orthodontic treatment;Patients who understood the protocol and could provide informed consent.

The inclusions criteria considered in the control group were as follows:I skeletal class, ANB = 2°/4°;Evidence in an extra-oral frontal photo of a media smile line, with a low line of the superior lip to third cervical gingival of superior central incisors with 1 mm of gingival exposed.The exclusions criteria for the study and control groups were as follows:Non-cooperative patients;Patients with previous orthodontic treatments;Inoperable patients;Patient with inadequate photo documentation;Patients with systemic pathologies;Patients in drug therapies.

All of the patients and their parents were informed with a written consent form about the scientific study and its potential benefits. 

The final sample consisted of 60 patients in the study group (30 female and 30 male) and 60 patients in the control group (32 female and 28 male).

For the study group, the extra-oral frontal photo was used to evaluate the amount of gingival maxillary exposure during a full smile as the distance between the lower edge of the upper lip and the gingival margin of the upper incisors (Figure 1). Dental casts were used to measure overjet, overbite and mesiodistal width of the crown of the upper central incisors to the dental equator with digital callipers (Figure 2). 

To quantify the amount of maxillary gum, a proportional ratio was made between millimetre width of the crown of superior central incisors and the pixel extension of this distance, obtaining a conversion coefficient called Photographic Gingival Exposure (PGE). The software used for this study was *Adobe Photoshop [photoshop, 2018 (19.0)]*.

Lateral cephalometric radiographs of all the subjects were calculated using 6 angular measurements, 4 linear measurements and 1 ratio between linear measurements. The angular measurements were:U1 to FH (upper incisors–Frankfort plane);U1 to PP (upper incisors–Palatine plane);U1 to NA (upper incisors–Nasion/A point);FMA (Frankfort plane–Gonion/Menton line);SN to PP (Sella-Nasion plane–Palatine plane);ANB (maxilla–mandibular relation, performed for only study group).

The linear measurements were:Overjet (millimeter sagittal distance between Incisor Superius and Incisor Inferius);Overbite (millimeter vertical distance between Incisor Superius and Incisor Inferius);U1-PP (the perpendicular length of a line dropped from U1 to the palatal plane);Is–Sts (millimeter distance between Incisor Superius and Stomion Superius);Sn–Sts (millimeter distance between Subnasale and Stomion Superius).

The ratio between linear measurements was as follows:(S-Go): (N-Me) = Posterior facial height/Anterior facial height.

### Statistical Analysis

The Data Analysis Extension of Microsoft Excel was used for the analyses of the data. All recorded data were analyzed using a *t*-test, which was performed to evaluate the differences between the groups and compare the mean differences of each cephalometric parameter between the group with gummy smile and the group with no gummy smile. For each dependent variable, it was checked that these variables are normally distributed within each group and that the variances of the two groups are equal. In this study, statistical significance was set at *p* < 0.05. 

## 3. Results

Of the 60 subjects with gummy smile, 30 were male and 30 were female; of the 60 subjects in the control group, 32 were female and 28 were male. The mean and SD of age for the study group and the control group and the amount of gingival exposure for the study group are shown in Table 1 and Table 2.

The mean and SD of the cephalometric measurements (angular, linear and ratio measurements) for the study group and the control group are shown in Table 3.

In the subjects with gummy smile, the amount of visible gingiva was 3.14 mm and, as shown in Table 2, the maxilla–mandibular relation, as represented by the angle ANB and calculated for only the study group, had a mean of 4.01°, indicating a tendency to the second skeletal class. 

Table 3 shows that in the study group, the mean of U1^PF, U1^PP and U1^NA was 115.09°, 112.58° and 24.58°, respectively. This indicated more protruded upper incisors than the control group, in which the mean of U1^PP and U1^NA showed a slightly ratiocinated upper incisor (Figure 3 and Figure 4). 

The mean of SN^PP and FMA, used in this study to indicate skeletal divergence, showed values within the normal range in both groups. 

With regard to linear measurements, the mean of overjet and overbite in the study group was 3.98 and 3.72 mm, respectively, which was significantly larger than in the control group. (Figure 5 and Figure 6).

Upper anterior dentoalveolar height, represented by U1-PP, was greater in the study group in comparison with the control group (Figure 7).

Sn–Sts, used in the present study to define the length of the upper lip, had a mean of 19.64 mm in the study group and 21.1 mm in the control group, indicating for both groups a reduction in the naso-labial distance from normal values.

The exposure of the upper incisors, indicated by Is–Sts, showed a significant increase in the study group compared with the control group (Figure 8).

## 4. Discussion

In November 2021, research was performed on PubMed (MEDLINE) to identify articles in the literature that investigated the existence of a specific craniofacial pattern in individuals with GS. The research was conducted by using the keywords: “gummy smile”, “craniofacial features”, “gingival display” and “dento–labial parameter” and aimed to select articles that were published in the last ten years. There was an insufficient number of articles on this topic, particularly in the time frame of our interest. Instead, more articles investigating this association have been published previously. 

In 1974, Singer [14] proposed the existence of characteristic craniofacial patterns associated with gummy smile. The author analyzed pre-treatment and post-treatment lateral cephalograms of 110 Caucasian females with gingival display and found an association between upward-tilted palate, high maxillary height, alteration of the maxillary incisors vertical position, short upper lip and gummy smile. This research found a statistically significant correlation between gingival exposure during smiling and increased positive torque of maxillary incisors with the FP, PP and NA lines as a reference in the GS group. Hayani et al. [11], conducted a study on a sample of 20 Syrian females with GS and did not find a statistical association between the gingival display and increased U1^Na. Wu et al. [15] reported upper incisors inclined labially with respect to the SN plane in patients of both sexes with GS. On the contrary, Sabri [16,17] suggested that proclined maxillary incisors can even reduce their exposure at rest. In the study group, the mean values of ANB were 4.01 degrees and indicated a skeletal class II relationship in subjects with gingival display, in agreement with Wu et al. In fact, in their study conducted on two hundred twenty-eight adolescents, more than one-half showed skeletal class II malocclusion, while no one patient exhibited a skeletal class III malocclusion [9,15]. Barbosa et al. [18] confirmed the strong influence of class II sagittal discrepancy on gummy smile.

This study found statistically significant correlations between the gingival display and increased values of overjet and overbite with a larger vertical distance between the Incisor and Stomion superior. These findings are similar to those from previous studies; Wu et al. [15] and Peck et al. [19] detected a significant correlation between excessive gingival display and larger values of overjet and overbite. In particular, Peck et al. [19] reported increased mean values of 1.5 mm and 1.0 mm of overjet and overbite, respectively, in subjects with GS compared to subjects who did not show gingiva when smiling, which is in accordance with previous studies [20,21].

Similar results were found by Khan et al. [22]; in their study, participants with GS had mean overjet and overbite of 3.15 and 3.03 mm, respectively, and, in particular, female subjects with gingival display had increased values of overjet compared to male subjects, while no gender differences were detected relating to overbite. On the contrary, Barbosa et al. [18] did not find a statistically significant correlation between GS and increased overjet in their study.

In this study, the inclination of the palatal plane to the cranial base (SN-PP) showed similar values to normal in both groups, and this result was the same as that obtained from other previous studies, such as Hayani et al. [11], Peck et al. [19], Singer [14] and Wu et al. [15] and confirms that the slant of the palatal plane was not a crucial factor in the evidence of gingival display.

With regards to anterior maxillary height (U1-PP), this study found a statistically significant correlation between this parameter and gummy smile. This is in agreement with previous studies by Peck et al. [19], Mackley et al. [23] and Ezquerra et al. [24], in which excessive gingival display and anterior dentoalveolar protrusion result clinically in gummy smile, and is also in agreement with the study by Hao Wu et al. [15].

This research showed a decrease in posterior/anterior facial height in the study group in comparison with the control group, but there was no significant correlation between gummy smile and reduced values of this ratio. This result was in disagreement with studies by Hayani et al. [11] and Wu et al. [15]: in their research, the posterior/anterior facial height in subjects with gummy smile showed a statistically significant decrease compared with the control group, indicating the presence of “long face” and vertical growth patterns in patients with gummy smile.

A vertical skeletal relationship was indicated in this study by FMA angle (Frankfort plane to Gonion Menton line); this angular measurement has normal values in both groups, indicating a normal growth pattern, unlike the results of the study by Hayani et al. [11], in which was found a vertical growth pattern in subjects with gummy smile.

This study found a statistically significant correlation between a reduction in the length of the upper lip and gummy smile. In fact, subjects with gingival display showed a significantly reduced value of this parameter in comparison with the control group and the normal value.

Relative to the upper lip length, previous literature results are controversial; many authors [25,26] have claimed that there are no differences in upper lip length between subjects with gummy smile and subjects without it. In his research, Singer [14] found increased measurement in the length of the upper lip above normal in 70 female patients with gummy smile. Peck et al. [19] also found the same results, and Wu et al. [15] recorded normal values in the upper lip length in males and slightly longer in females with gummy smile.

On the contrary, Redlich et al. [27] and Miron et al. [28] considered a short upper lip to be one of the main contributing factors in the occurrence of a gummy smile, and Barbosa et al. [18], in their research, found a statistically significant differences in upper lip length between the study group and the control group, revealing in subjects with gummy smile short upper lip at rest and during smiling.

The different ages of the populations of these studies can be used to justify controversial results; in fact, an acceleration in lip growth can be observed around the peak of puberal growth [25,26,27,28,29,30,31]. However, this study has several limitations: firstly, low sample size; a larger sample is definitely needed to get more consistent results. Another limitation was the impossibility of performing in vivo measurements of the amount of gingival exposure and the use of a frontal photo during a full smile, which may not accurately show the amount of gummy smile.

## 5. Conclusions

In the present research, a correlation emerged between gingival exposure and the presence of alterations to incisal torque in the vestibular direction and the quantity of maxillary gingiva evident during the smile, which is correlated in particular to the Is–Sts distance, overjet and overbite. The most indicative data, therefore, relate to the vertical position of the upper incisors, in particular with respect to the upper lip and the sagittal position. In particular, this proves a dentoalveolar component in the origin of gummy smile in the subjects of the study group. These results are useful for the purposes of a possible orthodontic therapeutic solution to this problem with dental intrusion techniques and indicate the need for greater attention to interceptive therapy of malocclusion with excessive vertical and sagittal dentoalveolar development. The data concerning that upper lip length imply that the origin of gummy smile is also muscular; this data is necessary for evaluating upper lip repositioning surgical techniques in order to solve this unsightly feature and rebalance the components of the smile.

## Figures and Tables

**Figure 1 healthcare-11-00344-f001:**
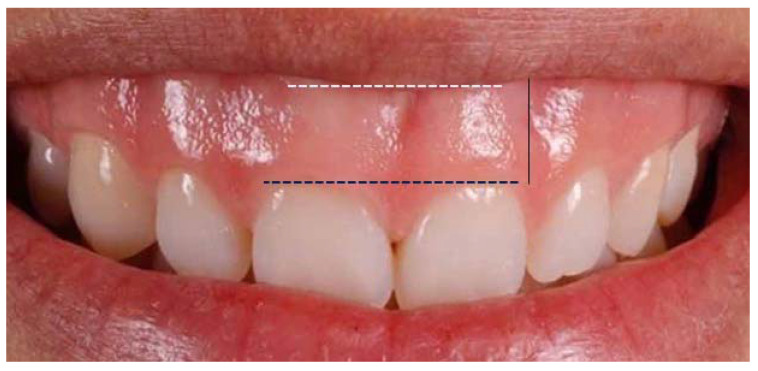
Gingival maxillary exposure during full smile evaluation.

**Figure 2 healthcare-11-00344-f002:**
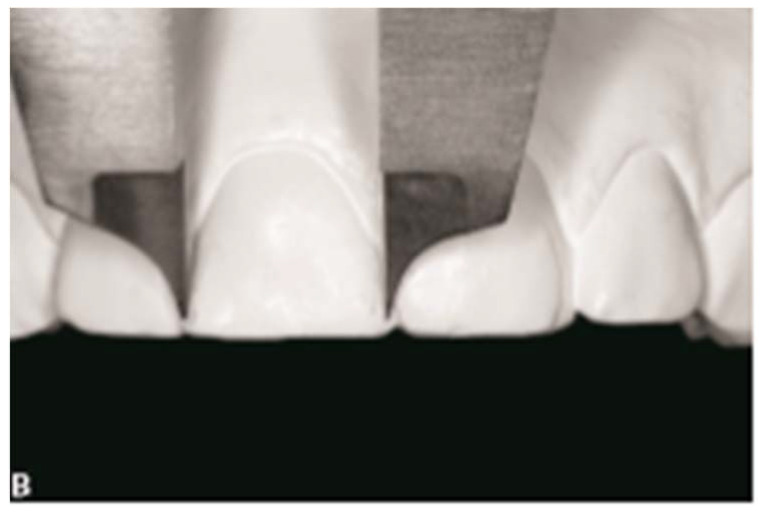
Dental casts evaluation.

**Figure 3 healthcare-11-00344-f003:**
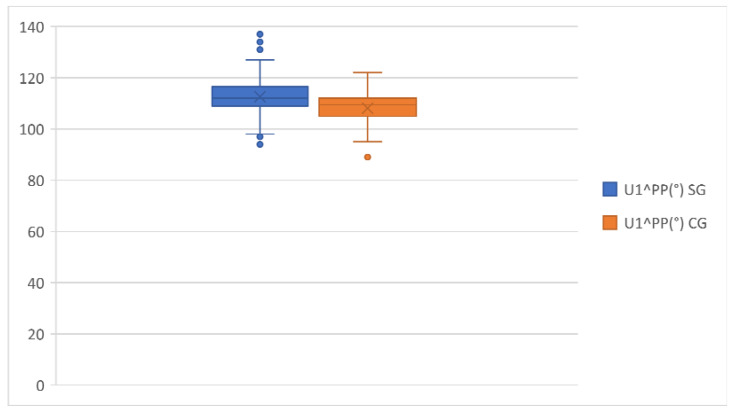
U1^PP comparison in the study group and the control group.

**Figure 4 healthcare-11-00344-f004:**
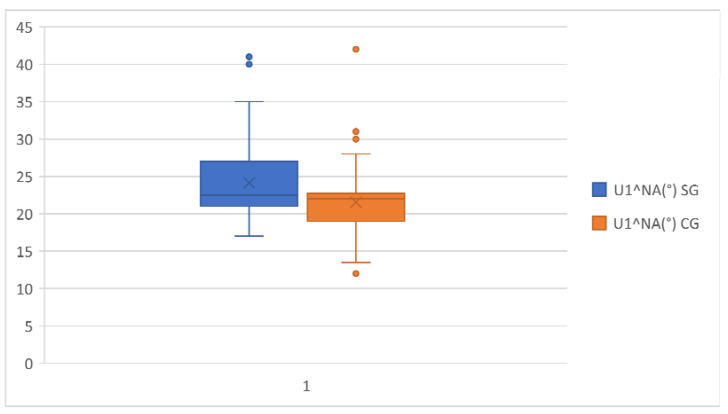
U1^NA comparison in the study group and the control group.

**Figure 5 healthcare-11-00344-f005:**
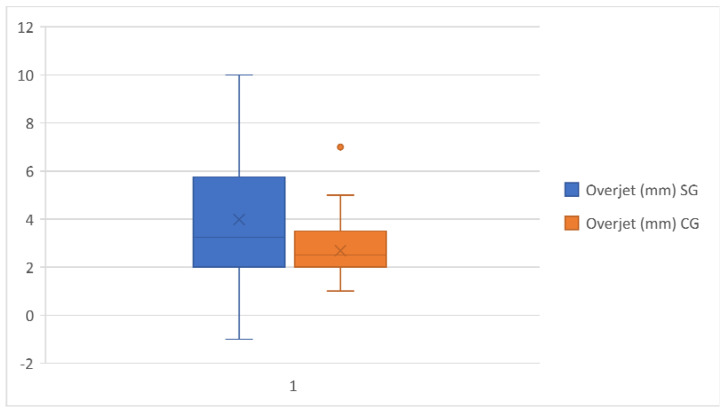
Overjet comparison in the study group and the control group.

**Figure 6 healthcare-11-00344-f006:**
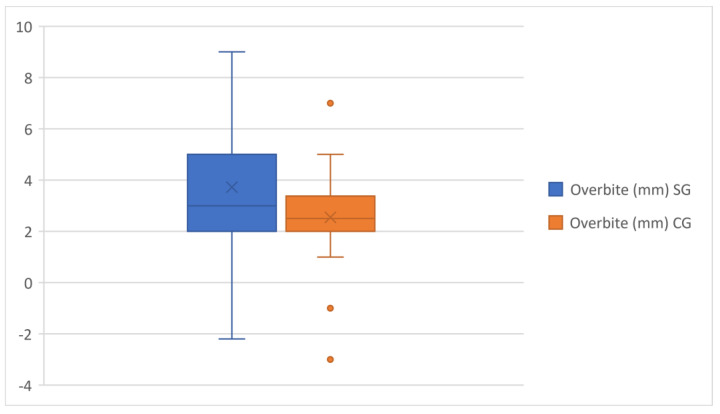
Overbite comparison in the study group and control group.

**Figure 7 healthcare-11-00344-f007:**
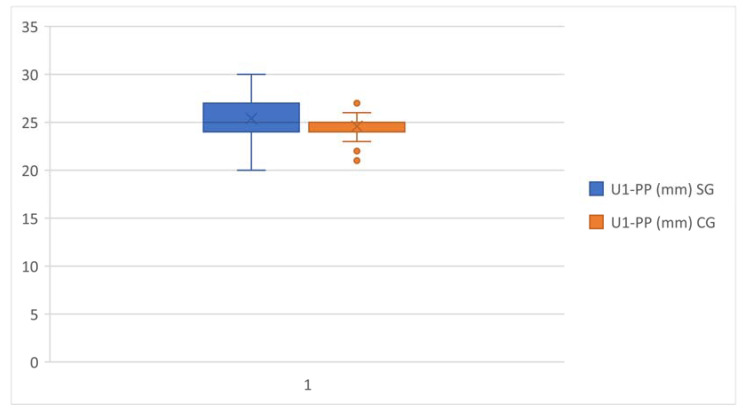
U1-PP comparison in the study group and the control group.

**Figure 8 healthcare-11-00344-f008:**
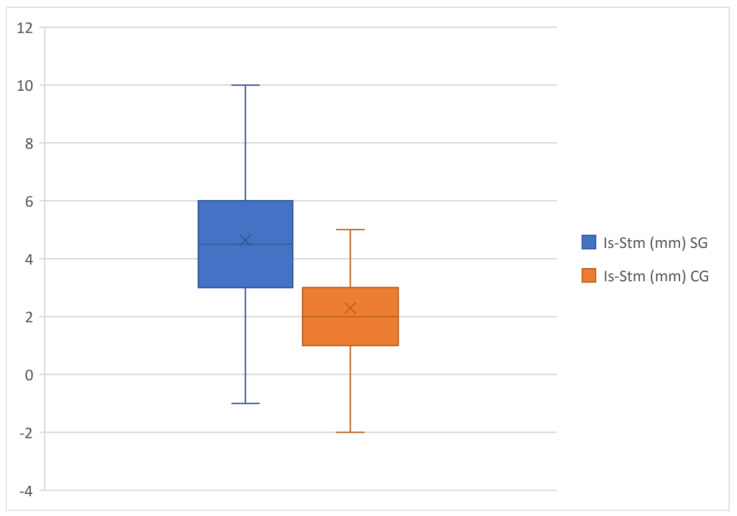
Is-Sts comparison in the study group and the control group.

**Table 1 healthcare-11-00344-t001:** Description of the samples in terms of age.

Group	Mean Age *(years)*	SD	Sample Size *(n)*
Study Group	13.2	4.12	60
Control Group	14.4	4.9	60

**Table 2 healthcare-11-00344-t002:** Gingival exposure in the study group.

Mean (mm)	SD	Sample Size *(n)*
3.14	0.74	60

**Table 3 healthcare-11-00344-t003:** Comparison of the angular (degree), linear and ratio measurements in two groups (Sig—Significance; * *p* < 0.05; ** *p* < 0.01; *** *p* < 0.001).

Variable	Study Group			Control Group			Significance	
	Mean	SD	*n*	Mean	SD	*n*	*p* Value	Sig.
U1 to FH (degrees)	115.09	8.76	60	110.52	5.19	60	0.0008	***
U1 to PP (degrees)	112.58	8.52	60	108.09	6.59	60	0.0017	**
U1 to NA (degrees)	24.58	11.9	60	21.54	4.72	60	0.0057	**
FMA (degrees)	27.88	5.67	60	23.97	5.88	60	0.0003	***
SN to PP (degrees)	8.29	1.77	60	8.23	2.25	60	0.8767	NS
ANB (degrees)	4.01	2.21	60	Not calculated			Not calculated	
Overjet (mm)	3.98	2.40	60	2.68	1.29	60	0.0004	***
Overbite (mm)	3.72	2.22	60	2.54	1.52	60	0.0010	**
U1-PP (mm)	25.38	1.827	60	24.58	1.201	60	0.0057	**
Is–Sts (mm)	4.63	2.39	60	2.30	1.45	60	0.0000	***
Sn–Sts (mm)	19.64	1.53	60	21.10	1.67	60	0.0000	***
(S-Go): (N-Me) (%)	60.07	3.82	60	61.66	3.94	60	0.0280	*

## Data Availability

No data availability.

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
