# Peer review of "Is There a Correlation between Gingival Display and Incisal Inclination in a Gummy Smile? Study on Cephalometric Parameters"

_healthcare, 2023, doi:10.3390/healthcare11030344_

Round 1

Reviewer 1 Report

This article is interesting but has two points that concern me and I believe should be addressed in depth:

First of all, has any Ethics Committee been passed? Because the patients underwent an orthodontic study, including X-rays, were they done for diagnostic indications? As for the control group, they were also X-rayed without needing it? So it is necessary to pass an Ethics Committee to approve such a study. If they have passed it, they must specify it in the paper.

Secondly, why were patients from 7 to 35 years old included? Justify this wide age range where growing and non-growing patients are mixed, where cephalometric values change.

Author Response

1) A retrospective study was carried used an archive, from wich patients were selected. They had gone to the Orthodontics UOC of the Department of Odontostomatological and Maxillo-Facial Sciences of the “Sapienza” University of Rome. These patients selected were well documented photographically and radiographically.

The recruitment’s period of time, mentioned in the manuscript, was the same lapse of time in which the archive searched for the patients, who could be selected for this study.

2) A literature review with the keywords specified in the manuscript was performed to evaluate any articlespublished in the last ten years in the literature on the topic.

3) Patients with systemic pathologies were excluded from ourstudy because of various pathologies, such as diabetes, leukemia, HIV, Chron’s disaese, lymphomas and varioustypes of vitamin.  These can determine gingivalhyperplasia or may require drug intake that can induce gingival swelling. In this way, the selection criteria of thisstudy could be altered.

4) I specified about the distribution of the variables in the Statistical analysis, at rows 128-130.

5) Sex and age of patients either of the Study group or the Control group were not the result of an intentional choice. Patients selected from the previously mentioned archivemet the same inclusion criteria of the study.

6) I included p-values at the Table 3.

Reviewer 2 Report

I think is an interesting manuscript, presenting an interesting and relatively well done research, which I think is not adequately presented/written.

The abstract should be rewritten. For example, study aim/objective would be better to be stated in the end of background section instead of Material and methods section.

In my opinion, the sections of the full text manuscript (introduction, method, results, discussion, conclusions) have some mixed information in regard to what they should contain. For example:

- study aim/objective should be stated in the end of introduction (it is stated somehow in the abstract, but I did not find it the full text manuscript)

- aspects as number of subjects included in the study, the sex distribution in study and control groups should be mentioned in the results section, not method; something similar, as the age is mentioned in results section, as it should be

- study limitations are not mentioned in the end of discussion

Besides these writing deficiencies, there are some other issues in my opinion that should be clarified as:

- please explain why it is state that is a “retrospective case-control study”, and not a cross-sectional study (rows 13-14)

- how the literature search mentioned in rows 51-55 is related to this study method

- some aspects of inclusion and exclusion criteria should be revised or better explained, as to what systemic pathologies they referred to (row 81)

- even if it’s easy to understand that F means females and M means males (rows 85-86), I think abbreviations should not be used if are not explained in the text

- you reported using t-test, and reported mean and SD; I suppose this method of data analysis and presentation was chosen after analyzing that data was normally distributed; if so this should be mentioned in the Statistical analysis from method section

- sex and age were, by your results similar in study and control group; was this intentional, by matching the groups by age and sex?; if so this should be mentioned in the method section

-  you said you compared groups by t-test, but this information is missing in table 3,4,5; I think would be better to include p-values

- also table 3,4,5 could be one table, and I think it would be better, when applied to include units of measuements (e.g., degres, mm)

- figures 5-10 in my opinion are hard to understand; I would recommend choosing another type of graph

Generally, the research is interesting and has an appropriate design, but I would strongly recommend a better presentation of it, which would make it more clearly to understand.

Author Response

1) An ethic committee opinion was not required as the patient’s radiographic documentation of this study was collected from the patient’s archive. They were treated by the Orthodontic UOC of the Department of Odontostomatological and Maxillo – Facial Sciences of the “Sapienza” University of Rome for the resolution of the malocclusion they were affected.  The patients of the control group selected for the study underwent radiographic investigations because they presented malocclusion treated in our Department. Therefore, no patients underwent radiographic investigations unnecessarily.

2) Patients were selected from the archive based on the study criteria. In particular, we selected patients with a minimum of 7 years of age and a maximum of 35 years of age who met the required criteria because we found that this clinical alteration occurs more frequently in this age range.

Reviewer 3 Report

There are several limitations that reduce overall enthusiasm for the study. First, it is unclear Why the authors  took such an extensive range of years for a small number of respondents. If they have already taken such an extensive age range, the age should also be taken into account, and the gingival exposure in the study group according the age of the participants should be  monitored. Please calculate the required sample size for this study. Also, the distribution of the data should be calculate and based on the distribution the appropriate tests should be used. 

Figure 1 and Figure 2 are needless, because what they show is undoubtedly already  mentioned in the texts. In addition, the Figure 5 till 10 are not the best solution for  displaying the data. Participants code number is randomly selected and in this way Readers of the manuscript could get the impression that patient 1 in the study group is paired with patient 1 in the control group, patient 2 in the study group is paired with patient 1 in the control group. 

Furthermore, the appearance of this curve is entirely insignificant because if the patients were arranged differently, it would look different too. In addition Mean Sd or Minimum, Maximum and Median (depended on distribution of the data) is quite enough. 

Author Response

1) The sample size was calculated starting from a large and unknown population and in particular the following formula was used:

Sample size =

z = z-score

e = margin of error

p = standard of deviation

I specified about the distribution of the variables in the Statistical analysis, at rows 128-130.

2) I deleted Figure 1 e 2 and I modified Figure 5 till 10.

Round 2

Reviewer 1 Report

You should explain why an Ethics Committee was not passed within the article.  You should clearly state

 that the records were requested for diagnostic purposes not for the study itself.

Author Response

Dear Reviewer,

An ethic committee opinion was not required as the patient’s radiographic documentation of this study was collected from the patient’s archive. They were treated by the Orthodontic UOC of the Department of Odontostomatological and Maxillo – Facial Sciences of the “Sapienza” University of Rome for the resolution of the malocclusion they were affected. The patients of the control group selected for the study underwent radiographic investigations because they presented malocclusion treated in our Department. Therefore, no patients underwent radiographic investigations unnecessarily.

Reviewer 3 Report

I am not satisfied with how the author explained the overall size calculation in his answer. it must be based on the results between the two groups and

be based on Cohen's d coefficient. Namely, it is a standardized effect size for measuring the difference between two group means.

Author Response

Dear reviewer,

thanks for your response. i sent u a document written by our statysical doctor.
